# Automated Detection of In-Home Activities with Ultra-Wideband Sensors

**DOI:** 10.3390/s24144706

**Published:** 2024-07-20

**Authors:** Arsh Narkhede, Hayden Gowing, Tod Vandenberg, Steven Phan, Jason Wong, Andrew Chan

**Affiliations:** 1Research, and Innovation Program, Glenrose Rehabilitation Hospital, 10105 112 Ave NW, Edmonton, AB T5G 0H1, Canada; 2Department of Electrical and Computer Engineering, Donadeo Innovation Centre for Engineering, University of Alberta, 11th Floor, 9211 116 St NW, Edmonton, AB T6G 1H9, Canada; 3Department of Mechanical Engineering, Donadeo Innovation Centre for Engineering, University of Alberta, 10th Floor, 9211 116 St NW, Edmonton, AB T6G 1H9, Canada

**Keywords:** ultra-wideband (UWB), machine learning (ML), human activity recognition (HAR), activities of daily living (ADLs), patient monitoring, indoor localization, older adults

## Abstract

As Canada’s population of older adults rises, the need for aging-in-place solutions is growing due to the declining quality of long-term-care homes and long wait times. While the current standards include questionnaire-based assessments for monitoring activities of daily living (ADLs), there is an urgent need for advanced indoor localization technologies that ensure privacy. This study explores the use of Ultra-Wideband (UWB) technology for activity recognition in a mock condo in the Glenrose Rehabilitation Hospital. UWB systems with built-in Inertial Measurement Unit (IMU) sensors were tested, using anchors set up across the condo and a tag worn by patients. We tested various UWB setups, changed the number of anchors, and varied the tag placement (on the wrist or chest). Wrist-worn tags consistently outperformed chest-worn tags, and the nine-anchor configuration yielded the highest accuracy. Machine learning models were developed to classify activities based on UWB and IMU data. Models that included positional data significantly outperformed those that did not. The Random Forest model with a 4 s data window achieved an accuracy of 94%, compared to 79.2% when positional data were excluded. These findings demonstrate that incorporating positional data with IMU sensors is a promising method for effective remote patient monitoring.

## 1. Introduction

Aging in place continues to be increasingly attractive for older adults. By 2047, the population of seniors aged 85 and older in Canada is expected to triple to 2.5 million [1]. As medicine and technology advance, managing physical and cognitive decline in increasing older populations is a significant challenge [2].

While long-term care has been a keystone to maintaining quality of life for older adults, its limitations are clearer than ever. Long-term care wait times range from 3 to 12 months in Alberta, BC, and Ontario [3,4]. The number of working Canadians per senior has fallen from 7.7 in 1960 to 3.4 in 2022, putting financial pressure on public funding for long-term care [1,5,6]. In Canada, a comprehensive set of 117 criteria has been established to benchmark quality of care, of which only 25 out of 117 (21.4%) criteria have been implemented in all provinces as of December 2022 [7]. Mental health is similarly worse in long-term care [8,9,10,11]. A study in Canada found that, among 50,000 seniors in long-term-care homes, 44% were either diagnosed with or had symptoms of depression [12]. Aging in place allows older adults to remain in their own home as they age while receiving physical, cognitive, social, or mental health support that can aid in the completion of activities of daily living (ADLs), such as eating, dressing, cooking, or housework [13,14,15,16].

While the current standard for assessing ADLs involves questionnaire-based assessments [17], a range of technologies have been developed to measure and support functional activities. Human activity recognition has exploded as a research field due to the rapid commercialization of wearable devices and improved computing power, making data analysis for detecting and classifying activities more practical [18]. Kinematic sensing technologies (accelerometers and gyroscopes), physiological sensors (photoplethysmography and piezoelectric respiratory sensors), video processing techniques, and wireless signal sensors (Wi-Fi, Bluetooth, and RFID) have all been researched to classify human activities [19].

However, each modality has specific areas of effectiveness. Kinematic sensors are powerful for posture and gesture recognition but suffer from drift when used in the long term [18]. Camera-based methods have robust classification ability and provide comprehensive information but are invasive of privacy and are resource intensive, although new technologies like depth cameras can reduce this concern [20]. Physiological sensors are focused on health-related outputs and not tailored to detect functional activities [21]. Wireless signaling technologies can maintain user privacy and provide some room-level localization information to detect frailty. Wireless signal technologies like Bluetooth beacons or passive infrared (PIR) sensors that maintain user privacy have shown how detecting room transitions can distinguish varying degrees of frailty [22,23,24].

Still, current technologies are not highly suited for detecting and eventually assessing functional activities over a long period of time. Ultra-Wideband (UWB) localization provides high-accuracy localization using radio waves to communicate and triangulate the user’s position [19]. To determine an individual’s location with a UWB system, a tag is worn that communicates with UWB anchors mounted around the perimeter of the room. UWB offers high accuracy within a range of less than 40 cm, compared to the 1–3 m offered by other technologies [25,26]. While most studies have focused on UWB radar for gesture and activity recognition to limited success [27,28,29], the combination of UWB as a localizer with accelerometry to classify functional activities has not been explored. The ability to combine high-accuracy localization and accelerometry data from a UWB wearable tag may allow for higher accuracy in the automated classification of activities of daily living.

Machine learning classification algorithms have been used in the past to detect ADLs using accelerometry data, infrared ambient sensors, cameras, temperature sensors, and hygrometry sensors [30,31,32]. Supervised machine learning through SVMs, Random Forests, and k-Nearest Neighbors are all designed with the goal of accurate classification, in this case, classification of activities of daily living. These models can accommodate complex multivariable data and can classify activities nearly in real time [33]. Pilot studies have evaluated the usage of Support Vector Machines, Random Forests, and k-Nearest Neighbor algorithms, resulting in 86–94% accuracy across these studies [30,31,32]. Further enhancements to the accuracy of activity classification could be achieved by including high-accuracy localization data.

High-accuracy localization data via UWB tags has not been thoroughly studied as a data source for human activity recognition. The ability to use furniture-level accuracy localization in combination with accelerometry in a low-profile wearable device provides an exciting prospect for detailed functional activity detection. This study evaluates the potential for high-accuracy localization with UWB tags to determine if the additional information provided by the tags enhances human activity recognition accuracy. Machine learning models were developed to incorporate localization data with existing wearable data from accelerometers and gyroscopes. The two aims of this study were to evaluate the optimal setup for using UWB real-time localization to provide high-accuracy position identification in a mock condo and to develop machine learning classification models that can categorize an individual’s ADLs based on UWB localization and Inertial Measurement Unit (IMU) sensor data.

## 2. Materials and Methods

### 2.1. Materials

A Pozyx (Pozyx, Ghent, Belgium) Creator Kit was used as the UWB technology in this study. The system included four to nine anchors mounted to the smart condo walls, a “Master Tag” connected to a computer, and a “Remote Tag” that is worn by the individual and includes both position and IMU data recording (Figure 1). The Creator Kit uses two-way ranging (TWR) for localization. The remote tag worn by the individual transmits low-powered radio waves to each of the anchors on the walls. The system analyzes the time of flight for the radio waves and tri-lateralizes the position of the tag in 3D space.

An Arduino Uno (Arduino, Ivrea, Italy) board was mounted to the Pozyx master tag and connected to a computer. An Arduino script provided by the manufacturer facilitated the calibration process. Bitrate was set to 850 kbit/s, the preamble length was set to 256, and the pulse repetition frequency was set to 64 Hz then tested for 1 min to ensure reliability. All the anchors and tags were set to communicate on the same channel using the same provided Arduino script.

The standard Python, version 3.11.0 (Python Software Foundation, Wilmington, DE, USA) script provided by the manufacturer was updated to output localization data in comma-separated value (CSV) format. The script collected and outputted the remote tag ID, the UNIX timestamp, and 16 sets of data at 16 Hz, utilizing the UWB localization system alongside the gyroscope and accelerometer sensors integrated into the remote tag worn on the individual’s wrist:Position in the ILS in X, Y, Z coordinates;Three-axis acceleration measured by on-board accelerometer (X, Y, Z directions noted in Figure 1);Three-axis linear acceleration with gravity removed from acceleration values;Three-axis angular velocity measured by on-board gyroscope;Euler angles calculated from gyroscope and accelerometry data (heading as rotation around Z-axis, pitch as rotation around Y-axis, and roll as rotation around X-axis in Figure 1);Barometric pressure (not used for any experiments in this study).

To simulate a real living environment, the experiments were conducted in the Independent Living Suite (ILS) at the Glenrose Rehabilitation Hospital (Figure 2). The ILS is a mock condo within the hospital, equipped with a fully functioning kitchen and bathroom. As seen in Figure 1c and Figure 2, anchors were attached to the walls throughout the ILS using temporary adhesives in an arrangement that maximized coverage around the entire condo. 

### 2.2. UWB Setup Evaluation

The first evaluation involved determining the optimal experimental setup for localization in the ILS to provide furniture-level location detection. A previous evaluation focused on anchor orientation, height differences, and various tag settings [26]. Two primary factors were tested in this study: the number of anchors placed in the space and the location that the tag was worn on the body. Furniture-sized locations in the ILS were measured with a tape measure, and each area was marked using masking tape placed on the floor (Figure 2). The respective coordinates of each corner for each location were then used to create virtual bounding boxes for labeling locations that would be used to calculate the accuracy of localization.

The anchors were spaced at intervals ranging from 2 to 20 m apart and placed at least 20 cm away from metal objects. All anchors were mounted vertically with the antenna at the top, facing outwards, and were positioned approximately 2 m above the ground. Placing all the anchors at similar heights deviated from the manufacturer’s recommendation of placing the anchors at varying heights for 3D positioning.

Remote tag testing involved only collecting position data from the tags. For the first factor, the remote tag was worn either on a lanyard or on a wrist strap on the right hand. The second factor was the number of active anchors used during the testing, starting at 5 and increasing to 9 anchors. Anchors were added in increasing order according to Figure 2. A total of ten combinations (two for body placement, five for number of anchors) were tested, each one tested five times. The combinations of furniture placement and body placement were randomized. The subject stood or sat within the bounding box of each of the locations for more than ten seconds, and the position of the UWB tag was recorded. 

The raw position of the tag was then compared to the preset coordinates of the piece of furniture to calculate accuracy. The data from the ten-second time series was then automatically labeled according to whether the coordinate was inside or outside the bounding box. Accuracy was calculated as the proportion of coordinate values in the time series that were within the bounding box as measured by the UWB.

### 2.3. Machine Learning Evaluation

The second experiment was conducted to investigate the ability of using the UWB-IMU system to classify ADLs. Supervised learning algorithms were trained to classify activities based on the data collected from the Pozyx Creator system. Nine activities related to Basic or Instrumental ADLs were selected for this purpose: walking, personal grooming/washing up, eating, wiping counters, standing (talking to a person), sitting (watching TV), loading/unloading the dishwasher, mopping, and changing clothes. These were selected in alignment with other studies on assessing activities of daily living [30,31,32]. The different variations in each activity explored in the trials are outlined in Table 1.

The subject performed 1 min of each activity in a different predetermined randomized order over 25 trials. Following the results of this paper’s first objective, a remote tag was worn on the individual’s dominant (right) wrist, and 9 anchors were installed around the suite to obtain the highest positional accuracy. Videos were recorded for each trial, with on-screen Unix timestamps used to compare and synchronize with the collected timestamped UWB positioning data.

Each video was examined to record the start and stop times of each activity. These timestamps were then used to label the collected data according to the corresponding activity. Data falling outside of these timestamps was discarded. The labeled positioning data were smoothed using a moving average with a window of 20 to eliminate noise. 

The cleaned and pre-processed data were then used to generate feature vectors using predefined time windows to approximate the continuous time-dependent data as a series of stationary points suitable for assigning activity labels and summarizing statistics over the specified corresponding time windows. Time windows ranging from 1 to 4 s were tested to determine the optimal window width for effective data capture in activity classification. Windows containing fewer than half the number of data points expected in optimal windows of the same size were discarded due to their unreliability, which typically occurred only when the UWB system had an error. Extracted features from each window included the mean, median, mode, maximum, minimum, and standard deviation of the signals obtained from the UWB system. 

Additionally, the count of peaks exceeding a specified threshold in the acceleration data, along with information on the room (according to room names in Figure 2) where the activities occurred, retrieved from the position data, was added to the data. The room name was labelled by collecting the location of the tag and comparing it with the predetermined room boundaries.

A total of 94 features were extracted for training the machine learning models. To simulate different systems and to explore the correlation between different input features and the model’s classification performance, four distinct data scenarios were chosen. The first data set comprised all available data. The second data set included only acceleration and gyroscopic data, totaling 75 features. A third data set contained only acceleration and position data, excluding gyroscopic data, resulting in 58 features. The final data set used only position data to simulate data collected from a UWB localizer with no additional sensors, resulting in 19 features. This is summarized in Table 2. Each data set was labeled, with the corresponding activity label serving as the final column. In total, 16 input feature vector files were generated, accounting for the combination of four different window widths and four distinct data scenarios.

For each feature vector, the location feature was represented using one-hot encoding, expanding the feature space to include as many columns as there were distinct locations in the data set, with values ranging from −1 to 1. Concurrently, the activity column was encoded using label encoding, assigning unique numerical labels to different activities while preserving the inherent ordinal relationship between activities. These encoded data were randomly split into a train–validation set, comprising 80% of the original data, and a test set, comprising the remaining 20% of the original data.

Five supervised machine learning models were trained using Python’s scikit-learn library: Polynomial SVM with degrees 2 and 3, Radial Basis Function SVM, k-Nearest Neighbor (kNN), and Random Forest (RF). As this is a pilot study focused on evaluating the utility of UWB-based localization data to enhance human activity recognition, more complex models including neural networks and gradient-boosted trees were not explored at this time. To optimize their performance and find the best overall model, a grid search approach was employed with the GridSearchCV library, exploring various combinations of hyperparameters. The train–validation set was used to train and evaluate the models using a randomized 4-fold stratified cross-validation approach. This involved dividing the train–validation set into four equal parts, where three parts, accounting for 60% of the original data, were used for training, while the fourth part, representing 20% of the original data, was used for validation or evaluation. A stratified approach was adopted to maintain the proportional representation of samples from each class in the data. Table 3 presents the various hyperparameters evaluated for the corresponding models.

The total number of combinations assessed in the grid search were 480: 16 factors multiplied by 15 SVM hyperparameter combinations, 7 Random Forest hyperparameter combinations, and 8 kNN hyperparameter combinations. Model performance was evaluated based on the average accuracy across the four folds when evaluated on the validation data. Mean values of metrics, including the F-measure, recall, precision, training accuracy, and validation accuracy, were exported to an Excel file for further analysis.

Among these models, four models trained on four distinct data scenarios were identified that exhibited the highest validation accuracy. Subsequently, these models were then trained on the combined train–validation set and assessed against the final test set. The resulting models were saved as Joblib files for future use. The reported accuracy reflects each model’s performance on the test set. Mean values of metrics such as the F-measure, recall, precision, and test accuracy and confusion matrices for each model were collected. Lastly, the Friedman test and Nemenyi test were calculated for the top four performers to compare if there was a significant difference between each data scenario.

## 3. Results

### 3.1. UWB Setup Evaluation

Figure 3 displays the accuracy of position readings, comparing where the tag was worn with the number of anchors. The toilet location always had an accuracy of 0% and was removed from the broader analysis. The nine-anchor configuration had the highest accuracy, with the wrist-worn tag out-performing the chest-worn tag in each case. There was a progressive increase in accuracy with more tags for the chest-worn tag, while for the wrist-worn tag, accuracy was interestingly highest when using either five or nine anchors.

The detailed analysis on the wrist-worn tag when using six and eight anchors showed a reduction in accuracy at the oven and fridge when adding the sixth anchor (Figure 2), dropping from 85% and 91% down to 46% and 44%, respectively. A similar reduction in accuracy occurred when adding anchor 8, with the bedroom chair location and bathtub dropping in accuracy from 95% and 96% down to 40% and 25%, respectively.

Figure 4 displays the furniture accuracy of the nine-anchor system against where the tag was worn, indicating reduced accuracy affecting primarily the bathtub and bedroom chair for the wrist-worn tag and reduced accuracies in the bathroom, desk, and bedroom chair for the chest-worn tag. Based on the data, the nine-anchor configuration was selected alongside the wrist-worn tag for future studies.

### 3.2. Machine Learning Activity Classification

Data collection consisted of 25 trials, each following a predetermined randomized order for the nine ADLs. Table 4 shows the aggregated quantity of data corresponding to each ADL across 25 trials after data cleansing, smoothing using a 20-row moving average window, and labeling procedures.

Each ADL has approximately 25 min of data, covering 25 trials with data collected at 16 Hz. However, the washing-up activity had reduced data (20 min) due to measurement interference from infrastructure within the washroom. Figure 5 shows a sample of the accelerometry, gyroscope, and position data for each of the nine activities over 3 s windows.

Activities with high accelerometry variability (up to 1.0 g) included changing clothes, wiping tables, loading the dishwasher, and washing up, while activities with moderate variability (up to 0.5 g) included standing still (talking), walking, eating, and watching TV. The gyroscope data changed most significantly when standing still (talking), changing clothes, and wiping tables, showing a high degree of changes in orientation of the wrist when performing these activities. Position changed moderately when loading the dishwasher and when mopping and changed significantly when walking but did not change for the other activities. 

The combinations of these three types of data are useful for specifying activities. For example, while loading the dishwasher and changing clothes have highly variable IMU data, their location data provide clarity on what activity is being completed. Conversely, wiping tables and eating were at the same location and could be distinguished by their accelerometry data. Mopping and walking both have moderately variable accelerometry data, but the distance moved helps to distinguish between them.

Model training was executed on 16 different types of feature vectors using a stratified 4-fold cross-validation approach. The goal was to identify the best model based on the mean accuracies across the four folds in a grid search. 

For a data scenario consisting of all data with a 1 s window (most amount of data points), both Polynomial SVMs of 2nd and 3rd degree showed an increasing trend in training time with increasing C values. Each model fit and evaluation took 1.5 s when C = 1 and 4.8 s when C = 1500. In contrast, the RBF SVM for the same input took longer times with lower C values, averaging 2.5 s per fit and evaluation when C = 1. However, the times decreased or remained unaffected with increasing C, with an average of 2.8 s when C = 1500. The kNN models were significantly faster, with an average time of less than 0.1 s per fit and evaluation, regardless of the hyperparameters evaluated. Similarly, model fit and evaluation time for RF increased with the number of estimators. The average time ranged from 0.7 s per fit and evaluation for N estimators = 10 to approximately 57 s for N estimators = 800. For all models and input feature vectors, the entire grid search to find the overall optimal model took approximately 55 min to 1 h to complete on an ASUS ROG Strix laptop (ASUS, Taipei, Taiwan), a 3.3GHz AMD Ryzen 9 5900HX processor, and 16 GB RAM running Windows 11 (Microsoft, Redmond, WS, USA).

The mean accuracy results for each fold when evaluated on the validation data set for each model when trained on different data sets with 4 s windows are shown in Figure 6.

For all models, data scenarios, and window sizes, metrics including precision, recall, and F-measure followed the same trend as accuracy of the model when evaluated on the validation data set. The model trained on all input features consistently exhibited the highest performance, followed by the model trained on all features except gyroscopic data. The model trained on all features except positional data showed the lowest performance, while the model trained only on positional data ranked third.

Window sizes generally had a marginal impact on all the models’ performance except on data where position was excluded. Window size changed accuracy by up to 4% for the Polynomial SVM of 2nd degree, 3.5% for the Polynomial SVM of 3rd degree, and 2% for Random Forests. However, for training without position data, these values were 7%, 5%, and 4%, respectively. For kNN, no such exceptions were noted; however, it was observed that window sizes impacted performance by up to 4.5% based on different hyperparameter combinations. The impact was more noticeable when uniform weights were used to train the kNN models.

Considering hyperparameters, for the Polynomial SVMs of 2nd degree and 3rd degree, Figure 6a,b show that the models’ performance improves with increasing the regularization parameter (C), but the improvement from C = 1000 to 1500 is marginal. For the RBF SVM, Figure 6c shows that model’s performance improves with increasing C from 1 to 100, but marginally worsens when C increases from 100 to 1500. In Figure 6d, for the RF model, performance improves with an increase in the number of estimators until 50, after which it plateaus with marginal improvements up to 800. In Figure 6e, kNN performs best when using inverse distance weighting rather than uniform weights. Performance is only marginally affected as the number of neighbors increases.

The type of input data had the greatest impact on the models’ performance. In comparison, the type of model used, the various hyperparameters evaluated, and the window sizes all generally had minor impacts, typically under 5%.

For each data scenario, the model with the highest prediction accuracy on the validation data set, after tuning hyperparameters, was selected. These models were then retrained on the train–validation data set and evaluated on the test data set. Across all data scenarios, RF with window size of 4 s consistently demonstrated the highest accuracy. The results are summarized in Figure 6.

Figure 7 shows that the model trained on all input features exhibited the highest performance, outranking all others. Not having position data had the strongest effect in reducing classification accuracy (dropping by 14%), followed by not having IMU data (drop by 6%) and then not having gyroscope data (drop by 1%). F-measure, precision, and recall also followed a similar trend as the mean classification accuracy, and these metrics are summarized in Table 5.

A Friedman test was conducted using a significance level of 0.05 and f^2^ effect type, resulting in *p* < 0.05, indicating that the ranks between some treatments were statistically significant. The Nemenyi test showed significant differences in accuracy between the all-data category and all data excluding position (*p* < 0.05), while there were no statistically significant differences between other groups (all data vs. excluding gyroscope at *p* = 0.69, all data vs. position only at *p* = 0.3, and excluding gyroscope vs. position only at *p* = 0.69). This indicated that position was the critical variable in differentiating activities.

Comparing individual activities, Figure 8 displays the confusion matrices for all data compared with no position data.

In the first two cases, mopping and washing up in the washroom had the lowest accuracy (88% and 91% including all feature vectors and 84% and 89% excluding gyroscope). When excluding position, changing clothes, eating, and using the dishwasher had the lowest accuracy (57%, 74%, and 77%, respectively), while for position-only sets for feature vectors, standing still and mopping had the lowest accuracy (70% and 80%, respectively). 

## 4. Discussion

### 4.1. Position Verification

Number of anchors and tag placement were both important for position accuracy. For the anchors, the Pozyx system automatically selects which anchors to use for tri-lateralization, which makes the relative position between the anchors and the localized position an important factor for accuracy. More anchors are not a guarantee of higher accuracy, as shown by the wrist-worn six-anchor configuration, which showed a drop in accuracy near the oven and fridge after an anchor was added in the kitchen. Future setups in residential homes will need to strategically place anchors with enough distance from commonly travelled areas to maintain tri-lateralization accuracy. 

Tag placement has a much stronger effect on accuracy, with a drop of up to 45% overall compared with anchor placement, which only changed the accuracy by 12% overall. The chest-worn tag was likely inferior because of interference from the individual’s body when the remote tag was close to the chest. UWB signals can be partially absorbed by water, reducing the signal speed, leading to reduced accuracy. This absorption may have contributed to lower accuracy near the bathroom sink and bathtub, due to plumbing interference within the walls of these areas. The toilet was particularly low performing, at 0% in all experiments, likely due to the combination of being on the edge of the capture area, being too close to an anchor, and significant water lines in the area.

### 4.2. Evaluation of Data Collection, Windowing, and Feature Vector Variables

From a data collection perspective, there were fewer data points collected in the washroom, despite spending the same amount of time in that space as in other spaces, due to data losses. As mentioned before, the water lines were a likely factor in reducing data reliability in these areas by 20%. UWB utilizes a wideband of radio frequencies to communicate effectively, allowing for low-power signals and minimal interference; however, UWB has high absorption in water and reflectivity in metals, resulting in unreliable signals. A unique feature in the ILS is the usage of metal studs in the hospital, which would significantly worsen signal quality throughout the ILS. We would expect improved performance in a residential wood-framed dwelling.

Window sizes had minimal impact on the models’ performance, generally changing accuracy by less than 4%, with exceptions when training models on data excluding position data. Still, the best-performing model in all scenarios used a window size of 4 s, which might provide an optimal balance between capturing enough context for accurate recognition and avoiding excessive irrelevant data.

Training models on different feature vectors showed that models trained with both IMU and location data can successfully distinguish between activities, achieving a 94.0% accuracy. Excluding gyroscopic information achieved similar accuracies at 93.0%, showing that gyroscopic data may not be important for activity classification. Conversely, models trained without positional information performed the worst, with an accuracy of 79.2%. Positional information was shown to improve accuracy when differentiating actions that have similar IMU outputs. Finally, models trained solely on positional data had an accuracy of about 87.8%, showing that location alone provides powerful information on what activity is being performed, whether using appliances in the kitchen, eating at various seats at the dining table, or performing personal grooming in the washroom.

For each model, precision and recall followed a similar trend to the model’s accuracy, indicating that the data are well balanced without bias towards one class or another. Models trained on both location and IMU data achieved precision and recall rates of around 94%, but when trained without positional data, precision and recall dropped to 80.3% and 79%, respectively, indicating an increase in both false positives and false negatives. 

### 4.3. Evaluation of Machine Learning Algorithms and Activities

Among the various models evaluated, RF proved to be the best-performing model across all scenarios. Although the SVM and kNN models showed promising performance, they did not match the performance of the RF model under any set of hyperparameter combinations. Notably, the optimal number of estimators for the RF model varied across different scenarios, indicating that no single value for this hyperparameter was universally the best, though the variation remained within 3%. 

Activity detection was consistently high when all feature vectors were available, with poorer accuracies in the washroom potentially due to interference in UWB data, and mopping being lower at 88%, likely due to the high variation in both gestures and locations from the mopping activity. When positional data were unavailable for training, the classification accuracy of standing still and eating decreased as other activities produced similar gyroscopic and accelerometry data. For position-only data, mopping performed relatively worse than all other activities, having the highest variation in locations that could overlap with other locations.

The statistical analysis using the Friedman and Nemenyi tests showed that there was a significant difference between using all data types to develop the optimal model and removing position as a data source. The high-accuracy position appeared to have the strongest effect on classification accuracy, indicating its importance in helping to differentiate activities that may have similar accelerometry parameters (watching TV and eating in Figure 5) but may be conducted in different locations. A broader range of activities being performed in various locations need to be completed to better determine if functional activities can be differentiated with added position data, or perhaps if added context (where a person comes from or where a person goes to after performing an activity) can help identify what a person is doing.

### 4.4. Comparison to the Literature

This study evaluates the efficacy and advantages of using UWB technology for recognizing human activities in the context of remote patient monitoring. A systematic study found that IMU sensors within smartphones are well suited for HAR in health research [18]. Further investigation of 90 subjects revealed that RF models can classify the subsamples with almost 90% accuracy using these IMU sensors and smartphones [34]. The current study found an accuracy of 79.2% when including only IMU data but was able to outperform the study by Sikder et al. when including position data, showing that position data are a promising avenue.

In a previous study, it was found that the optimal window size, as determined by recognition performance, was dependent on the classification paradigm [35]. However, the findings of the current study, which used a 4 s window size, deviate from the previous study that found optimal window sizes as 1–2 s for ADL recognition. In this study, the duration of the activities captured would be typically greater than 30 s, making a 4 s window an adequate timespan to capture features of the activity. 

Many studies have also explored the use of UWB radars for activity recognition. A prior study found that UWB Doppler radars are a promising research avenue for smart homes. In that study, RF models achieved around 80% accuracy in identifying 15 different ADLs in a 40-square-meter apartment with 10 different subjects [27]. Our study only has one subject but found promising accuracies of 94% using a UWB tag-based system rather than radar. Another study that used the same UWB radar mounted on a mobile robot rather than as a wearable device achieved an impressive accuracy of 99.6% using deep learning models, specifically the Long Short-Term Memory (LSTM) model [36]. However, this study by Noori et al. focused on detecting postures rather than functional activities.

These findings are encouraging, as they demonstrate that UWB radars can effectively monitor patients without requiring them to wear any devices. Additionally, the high accuracy achieved with deep learning models, specifically RNN models like LSTM, indicate that this approach is a step in the right direction for further research and development in remote patient monitoring.

### 4.5. Strengths, Limitations, and Future Work

To our knowledge, this is the first study using UWB localization tags to perform human activity recognition in the home setting. This study had several strengths, including the use of the ILS to simulate a real-world home environment, being more applicable to the real world. The data collection process adopted a generalized approach to include various ways of performing the activities, adding reliability to this study. From a machine learning perspective, a grid search approach was used to identify the best model for each scenario with the optimal hyperparameter combination. Although this approach was costly in terms of time, it was effective for finding the most accurate models. A 4-fold stratified cross validation was used to ensure each fold maintained the correct class representation. 

Shortcomings in the data analysis included having only one participant within one setting, though most other studies also only use one setting at this time. Handedness was not evaluated in this study. This study only used simple supervised classification algorithms rather than algorithms typically used for time series data, including RNNs and LSTMs. More hyperparameters could have been evaluated, but accuracies were already reaching a plateau in the current study. The model would likely only be applicable in the ILS at this time. From an implementation perspective, the UWB anchors were mounted at similar heights, with no consideration for a multi-story environment. The current UWB setup uses two-way ranging that significantly reduces battery life. The tested system is for prototyping, not for implementation.

## 5. Conclusions

To our knowledge, this is the first study using IMU data with UWB localization to identify activities of daily living in a simulated home setting. Our machine learning model could classify functional activities with 88% accuracy using UWB localization data exclusively, and 94% when accelerometer and gyroscope data were included. For the common ADLs included in this study, UWB can be an effective tool in classifying the activities of older adults within the home.

This study is a preliminary step towards classifying ADLs within homes using novel localization techniques. To classify more complex ADLs and improve the accuracy of current classifications, more advanced deep learning models such as RNNs with LSTM, GRU, and possibly CNN models need to be trained. Additional activities and additional subjects would be needed alongside investigating more robust UWB sensors like the Pozyx Enterprise system with comfortable wrist-mounted trackers to extend its application to patient homes. Monitoring older adults remotely can support them in living independently at home and enhance their overall well-being.

## Figures and Tables

**Figure 1 sensors-24-04706-f001:**
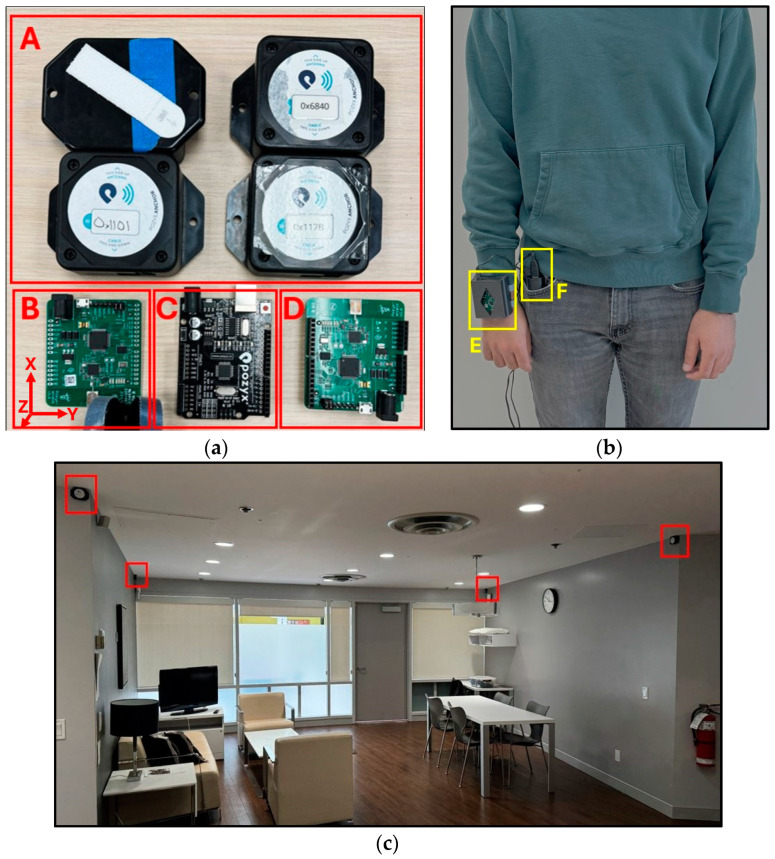
(**a**) Creator Kit components indicated by boxes around components and labeled A–D. A: Anchors with anchor IDs, B: Pozyx remote tag, C: Arduino Uno, D: Pozyx master tag; coordinate system for X, Y, Z is shown in B. (**b**) Demonstration of wearing the remote tag highlighted by boxes and labeled E and F. E: wrist-worn Pozyx remote tag inside custom 3D-printed case, F: battery pack. (**c**) Living room anchor setup indicated by boxes around the anchors.

**Figure 2 sensors-24-04706-f002:**
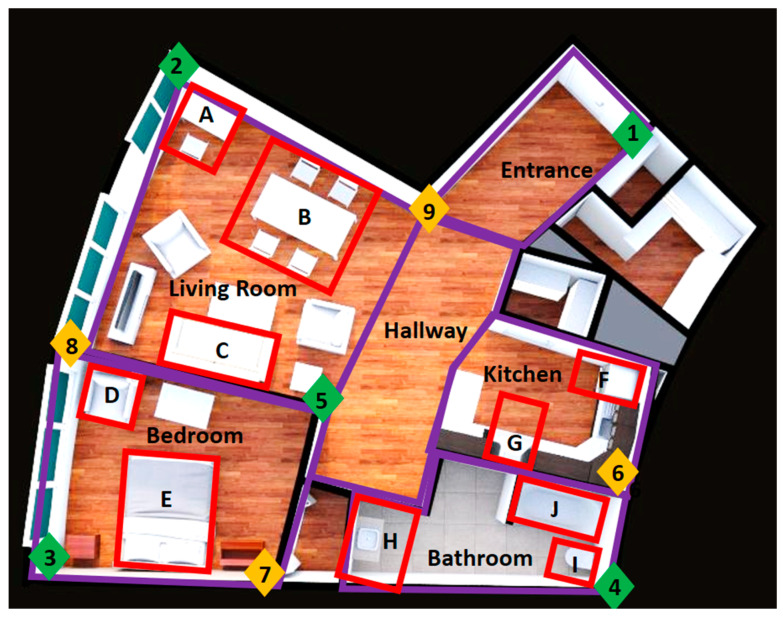
Independent Living Suite setup including furniture locations indicated by boxes around furniture and labeled A–H. Furniture labels: A—desk, B—dining table, C—couch, D—bedroom chair, E—bed, F—fridge, G—oven, H—sink, I—toilet, J—bathtub. The anchor locations are indicated by diamonds and labeled from 1–9. Anchors 5 and 8 are located on the living room side of the wall. Room locations are bounded by purple boxes, splitting the condo into six sections: living room, bedroom, bathroom, hallway, kitchen, and entrance.

**Figure 3 sensors-24-04706-f003:**
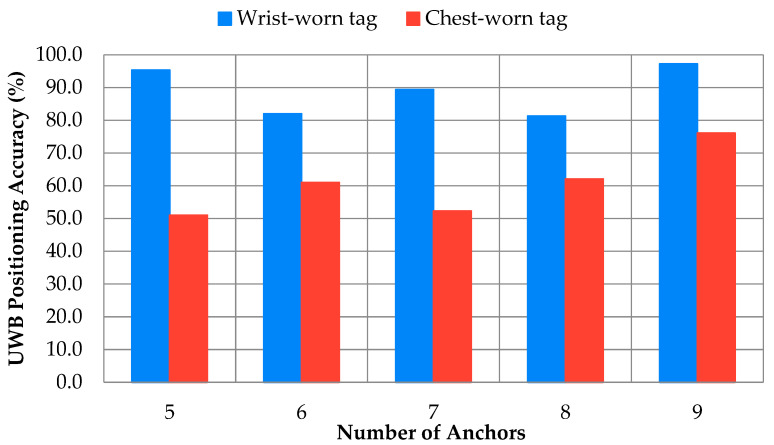
Positioning accuracy against number of anchors, differentiating wrist-worn tag from chest-worn tag.

**Figure 4 sensors-24-04706-f004:**
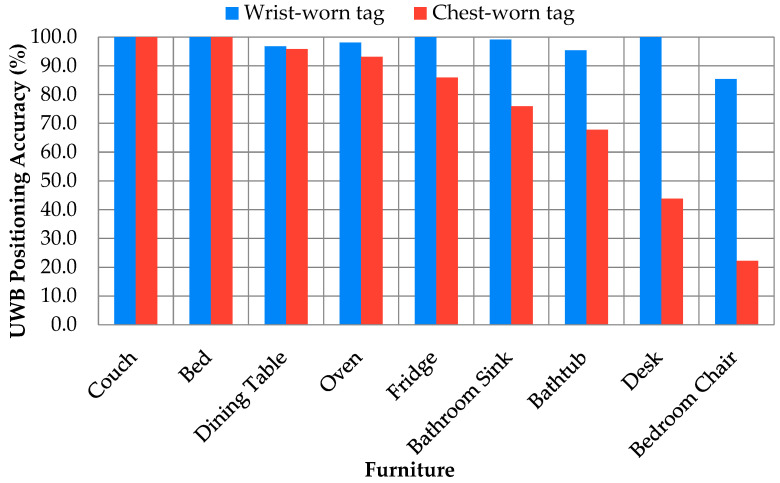
Positioning accuracy of furniture placements for wrist- vs. chest-worn tag for 9-anchor configuration.

**Figure 5 sensors-24-04706-f005:**
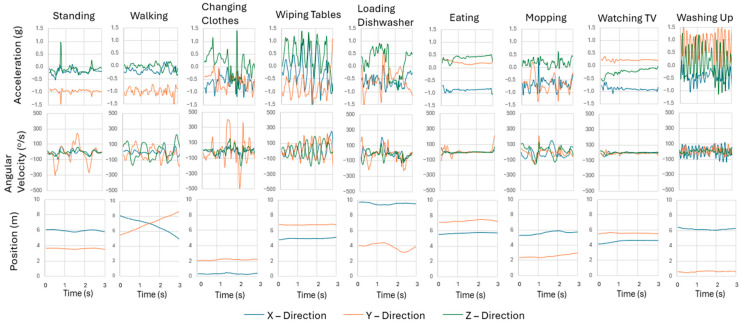
Sample of raw data from accelerometry, angular velocity, and position data from wearable tag. Acceleration included 3 directions and gravity. The gyroscope included 3 directions, while position included only the X or Y coordinate within the Independent Living Suite.

**Figure 6 sensors-24-04706-f006:**
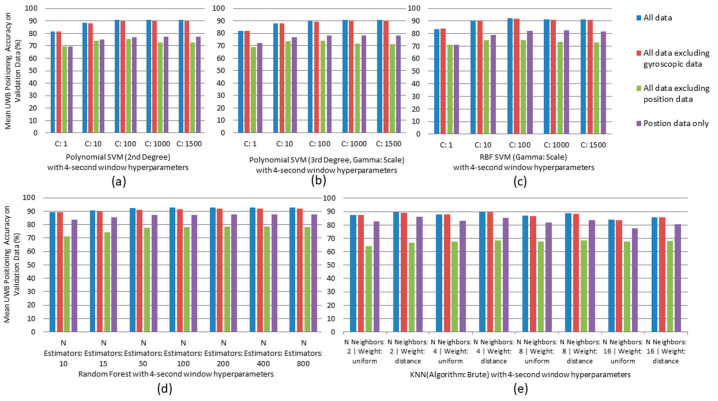
Models’ mean accuracy when evaluated on validation data set with 4 s windows for (**a**) Polynomial SVM with degree 2, (**b**) Polynomial SVM with degree 3, (**c**) RBF SVM, (**d**) RF, (**e**) kNN.

**Figure 7 sensors-24-04706-f007:**
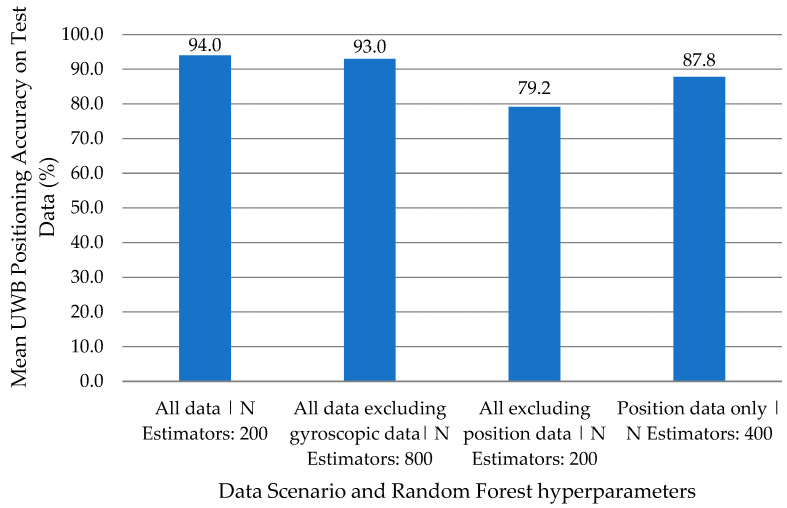
Best models (Random Forest with 4 s windows) and their mean accuracy when evaluated on test data set for different data scenarios.

**Figure 8 sensors-24-04706-f008:**
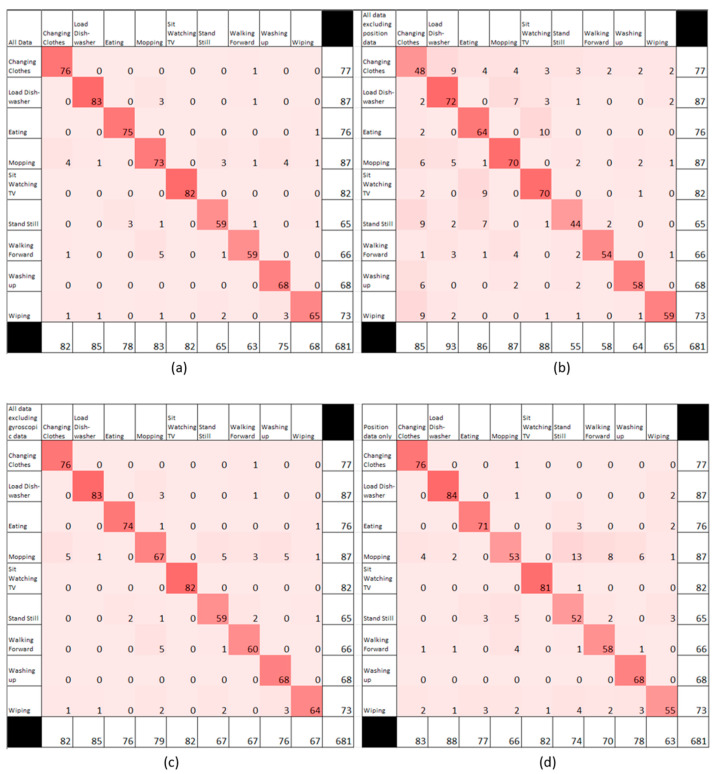
Confusion matrices for (**a**) all feature vectors (IMU and position) and (**b**) excluding position only, (**c**) excluding gyroscope only, and (**d**) including position only. Level of red shading indicates magnitude of the number in the cell.

**Table 1 sensors-24-04706-t001:** Variation in activities explored for each activity of daily living.

Activity of Daily Living	Variations in Actions Explored
Changing clothes	Dressing and undressing in button-up shirts, jackets with zippers, round-neck sweaters, pants, and shorts.
Dishwasher	Opening the dishwasher with the subject facing towards or away from it; loading and unloading plates, glasses, forks, knives, and cups.
Eating	Simulating eating spaghetti with a twisting action of a fork, eating burgers with hands, eating chicken wings with hands, eating soup with a spoon, drinking beverages, taking breaks to simulate chewing, and eating with a knife and fork.
Mopping	Simulating dry and wet mopping, performing linear and circular mopping actions.
Sitting (watching TV)	Sitting in various postures while watching TV and in different positions on the couch.
Standing still	Standing in different parts of the ILS. Simulating speaking to someone, stretching.
Walking	Walking at varying speeds in a random manner throughout the ILS.
Washing-up	Simulating washing the face, combing hair, shaving, brushing teeth, applying cream/moisturizer, and wiping the face.
Wiping	Simulating linear, circular, and random dusting actions.

**Table 2 sensors-24-04706-t002:** Number of total features extracted for each data scenario.

	All Data	All Data Excluding Positional Data	All Data Excluding Gyroscopic Data	Positional Data Only
X, Y, Z Position (18 features from 6 extracted statistics each)	x		x	x
X, Y, Z Acceleration (18 features from 6 extracted statistics each)	x	x	x	
X, Y, Z Linear Acceleration (18 features from 6 extracted statistics each)	x	x	x	
X, Y, Z Angular Velocity (18 features from 6 extracted statistics each)	x	x		
Heading, Pitch, Roll (18 features from 6 extracted statistics each)	x	x		
Location in ILS (1 feature)	x		x	x
X, Y, Z Peak Acceleration (3 features)	x	x	x	
Total Input Features	94	75	58	19

‘x’ indicates that a particular feature was included in that data scenario.

**Table 3 sensors-24-04706-t003:** Hyperparameters evaluated for each model.

Model	Hyperparameters Evaluated	Hyperparameter Values	Number of Values
SVM	Kernel	Polynomial (2nd Degree), Polynomial (3rd Degree), RBF	3
	C (Regularization Parameter)	1, 10, 100, 1000, 1500	5
	Gamma (RBF, Polynomial SVM—3rd Degree)	Scale	1
Random Forest	N Estimators	10, 15, 50, 100, 200, 400, 800	7
k-Nearest Neighbors	N Neighbors	2, 4, 8, 16	4
	Weights	Uniform, Distance	2
	Algorithm	Brute	1

**Table 4 sensors-24-04706-t004:** Total quantity of data per ADL.

Activity	Data Points	Approximate Time (Mins)
Changing clothes	24,772	25.80
Dishwasher	21,324	22.21
Eating	26,047	27.13
Mopping	24,159	25.17
Sitting (watching TV)	25,848	26.93
Standing still (talking)	23,538	24.52
Walking	22,824	23.78
Washing-up	19,165	19.96
Wiping	23,702	24.69

**Table 5 sensors-24-04706-t005:** Model performance metrics per data scenario.

Data Scenario	Accuracy (%)	Precision Score (%)	Recall Score (%)	F-Measure (%)
All data	94.0	93.9	94.0	93.9
All data excluding gyroscopic data	93.0	92.8	93.1	92.9
All data excluding position data	79.2	80.3	79.0	79.4
Position data only	87.8	87.3	88.0	87.3

## Data Availability

Data is available upon request.

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
