# Peer review of "Automated Detection of In-Home Activities with Ultra-Wideband Sensors"

_sensors, 2024, doi:10.3390/s24144706_

Round 1

Reviewer 1 Report

Comments and Suggestions for Authors

This work studied on human activity recognition based on Ultra-Wideband (UWB) and Inertial Measurement Unit (IMU)  signal data.The researchers gathered their own dataset using UWB and IMU sensors. They then applied machine learning algorithms to this dataset in order to develop models capable of recognizing and categorizing various human activities. To enhance the quality and clarity of this paper, the authors are advised to consider and address the following feedback and recommendations.

1) The authors should elaborate on the rationale for employing machine learning models and the underlying mechanisms through which these models operate to address the identified challenges.

2) To improve the clarity and impact of the introduction, the authors should explicitly highlight the key contributions of their research.

3) To provide context and demonstrate the novelty of their work, the authors should include a dedicated section that discusses related studies and prior research in the field.

4) To gain a better understanding of the UWB and IMU datasets, the authors should present visualizations of representative samples and discuss the distinguishing features and characteristics that differentiate the various human activities within the data.

5) Add the Friedman test, Nemenyi test, and Wilcoxon test to the metrics in Table 5.

6) To effectively summarize the key findings, implications, and potential future directions of their research, the authors should include a dedicated conclusion section at the end of the paper.

7) Improve the language quality of the paper and include a table summarizing the acronyms.

8) As this study involves the collection of data from human subjects, the authors should provide information about the Institutional Review Board (IRB) approval or exemption obtained for their research, in order to demonstrate compliance with ethical guidelines and protect the rights and welfare of the participants.

Comments on the Quality of English Language

Minor editing of English language required

Author Response

Thank you for the helpful comments to enhance the clarity of our article. I’ve included the responses to each of the items here: 

Comment 1: The authors should elaborate on the rationale for employing machine learning models and the underlying mechanisms through which these models operate to address the identified challenges. 

We have added clarity on why machine learning models were used and how the classification models are appropriate for the challenge of classifying activities of daily living. We have included reasoning on why more complex models including neural networks and gradient-boosted trees were not used in this case: simply that this was an initial pilot study determining the utility of UWB and future studies will include evaluating neural networks and other models. 

Line 84-89: Machine learning classification algorithms have been used in the past to detect ADLs using accelerometry data, infrared ambient sensors, cameras, temperature sensors, and hygrometry sensors [25–27]. Supervised machine learning through SCMs, Random Forests and K-Nearest Neighbors are all designed with the goal of accurate classification, in this case, classification of activities of daily living. These models can accomodate complex multivariable data and can classify activities nearly in real-time. 

Line 244-246: As this is a pilot study focused on evaluating the utility of UWB-based localization data to enhance human activity recognition, more complex models including neural networks and gradient-boosted trees were not explored at this time. 

  

2) To improve the clarity and impact of the introduction, the authors should explicitly highlight the key contributions of their research. 

We have added clarification on the key contributions. This is the first study on UWB-based localization’s contribution to human activity recognition using machine learning. 

  

Line 94-101: High-accuracy localization data via UWB tags has not been thoroughly studied as a data source for human activity recognition. This study evaluates the potential for high-accuracy localization with UWB tags to determine if the additional information provided by the tags enhance human activity recognition accuracy. Machine learning models were developed to incorporate localization data with existing wearable data from accelerometers and gyroscopes. 

3) To provide context and demonstrate the novelty of their work, the authors should include a dedicated section that discusses related studies and prior research in the field. 

We have expanded our section on providing context and demonstrating novelty. Current human activity recognition technologies are limited by acceptability (privacy invasiveness in camera and microphone-based technologies), accuracy (IMU-drift and poor localization accuracy for Wifi and Bluetooth), and specific applications (physiologic sensors are not sensitive to functional activities). Ultra-wideband can overcome these drawbacks by protecting privacy while providing enough accuracy to give information on functional activities.  Ultra-wide-band localization has not been used for functional activity recognition and has great potential to become a powerful tool to assess and monitor if residents are doing self-care activities.  

Line 50-73: While the current standard for assessing ADLs involves questionnaire-based assessments [17], a range of technologies have been developed to measure and support functional activities. Human activity recognition has exploded as a research field due to rapid commercialization of wearable devices and improved computing power making data analysis for detecting and classifying activities more practical [18]. Kinematic sensing technologies (accelerometers and gyroscopes), physiological sensors (photoplethysmography, piezoelectric respiratory sensors), video processing techniques and wireless signal sensors (Wifi, Bluetooth, RFID) have all been researched to classify human activities [19].   

However, each modality has specific areas of effectiveness. Kinematic sensors are powerful for posture and gesture recognition but suffer from drift when used over the long-term [18]. Camera-based methods have robust classification ability and provide comprehensive information but are invasive of privacy and are resource intensive, although new technologies like depth cameras can reduce this concern [20]. Physiological sensors are focused on health-related outputs and not tailored to detect functional activities [21]. Wireless signaling technologies can maintain user privacy and provide some room-level localization information to detect frailty. Wireless signal technologies like Bluetooth beacons or passive infrared (PIR) sensors that maintain user privacy, have shown how detecting room transitions can distinguish varying degrees of frailty [22–24].  

Still, current technologies are not highly suited for detecting and eventually assessing functional activities over a long period of time. Ultra-Wide Band (UWB) localization is a high-accuracy localization technique that uses radio waves to communicate and triangulate the user’s position [19]. 

  

4) To gain a better understanding of the UWB and IMU datasets, the authors should present visualizations of representative samples and discuss the distinguishing features and characteristics that differentiate the various human activities within the data. 

An additional figure (Figure 5) has been added, showing visuals of the samples with brief comment on the distinguishing features that differentiate each human activity. 

Line 315-328: Activities with high accelerometry variability (up to 1.0 g) included changing clothes, wiping tables, loading dishwasher and washing-up, while moderate variability (up to 0.5g) included standing still talking , walking, eating and watching TV. Gyroscope data changed most significantly when standing still talking, changing clothes and wiping tables, showing a high degree of changes in orientation of the wrist when doing these activities. Position changed moderately when loading the dishwasher and when mopping, changed significantly when walking, but did not change for the other activities. 

The combination of these three types of data are useful for specifying activities. For example, while loading dishwasher and changing clothes have highly variable IMU data, their location data provides clarity on what activity is being completed. Conversely, wiping tables and eating were at the same location and could be distinguished by their accelerometry data. Mopping and walking both have moderately variable accelerometry data, but the distance moved helps to distinguish between them. 

  

5) Add the Friedman test, Nemenyi test, and Wilcoxon test to the metrics in Table 5. 

  

We have included Friedman and Nemenyi tests in the analysis. It wasn’t quite clear to us how the tests were intended to be applied to our particular data set, but we’ve applied them to the metrics presented in Table 5. We did not include a Wilcoxon which is recommended to have at least six datapoints to complete as we didn’t feel it would be a valid test in this case. Comment has been added below: 

Line 395-402: A Friedman test was conducted using a significance level of 0.05 and f2 effect type, resulting in P<0.05, indicating that the ranks between some treatments were statistically significant. The Nemenyi test showed significant differences in accuracy between the all-data category and all-data excluding position (P<0.05), while there were no statistically significant differences between other groups (all data vs excluding gyroscope at P=0.69, all data vs position-only at P=0.3, and excluding gyroscope vs position-only at P=0.69). This indicated that position was the critical variable in differentiating activities. 

Line 480-489: The statistical analysis using Friedmand and Nemenyi tests showed that there was a significant difference between using all data types to develop the optimal model, and re-moving position as a data source. The high-accuracy position appeared to have the strongest effect on classification accuracy, indicating its importance in helping to differen-tiate different activities that may have similar accelerometry parameters (watching TV and eating in Figure 5), but may be done in different locations. A broader range of activities being performed in various locations needs to be completed to better determine if func-tional activities can be differentiated with added position data, or perhaps if added con-text (where a person comes from, or where a person goes to after doing an activity) can help identify what they were doing. 

6) To effectively summarize the key findings, implications, and potential future directions of their research, the authors should include a dedicated conclusion section at the end of the paper.  

We have added a dedicated conclusion section summarizing the findings, and the implications and future directions of our research. We are excited to see the potential of UWB localization to provide rich information on functional activities. 

Line 542-555 This is the first study combining accelerometry with UWB localization  to our knowledge, demonstrating that in a simulated home setting, a machine learning model could classify activity with 88% accuracy using UWB localization data exclusively, and 94% when accelerometer and gyroscope data was included. For the common ADLs included in this study, UWB can be an effective tool in classifying the activities of older adults within the home.  

This study is a preliminary step towards classifying ADLs within homes using novel localization techniques. To classify more complex ADLs and improve the accura-cy of current classifications, more advanced deep learning models such as RNNs with LSTM, GRU, and possibly CNN models need to be trained. Additional activities and additional subjects would be needed alongside investigating more robust UWB sensors, like the Pozyx Enterprise system with comfortable wrist-mounted trackers to extend application to patient homes. Monitoring older adults remotely can support them in liv-ing independently at home and enhance their overall well-being.  

  

7) Improve the language quality of the paper and include a table summarizing the acronyms.  

A list of abbreviations has been added to the article at the end. The article has been edited additional times for its language quality. 

8) As this study involves the collection of data from human subjects, the authors should provide information about the Institutional Review Board (IRB) approval or exemption obtained for their research, in order to demonstrate compliance with ethical guidelines and protect the rights and welfare of the participants.  

We had verbally confirmed with our REB that this study did not require ethics approval but have now also confirmed with a written letter that an ethics approval was not required and this study is exempted. We have attached their exemption letter in our submission. 

Reviewer 2 Report

Comments and Suggestions for Authors

This manuscript showed fascinating results. However, the completeness of the research results can be further improved by revising the following items.

Q1) The authors increased the number of UWB tags from 5 to 9 to evaluate the accuracy of UWB location estimation. When increasing the number of UWB tags, were the tags installed in the order defined by the tag numbers in Figure 2?

Q2) The locations of Tag 5 and Tag 8 in Figure 2 are unclear. The specific locations of each tag, whether in the living room or the bedroom, need to be indicated.

Q3) Figure 3 compares the accuracy according to the location of the UWB anchors. However, the manuscript does not explain how the actual positions were recorded during the experiments and how the exact position information was synchronized with the UWB measurement data. The authors should describe this in detail.

Q4) What are the criteria for the Heading, Pitch, and Roll measurements defined in Table 2?

Q5) According to Figure 3, when the UWB is located on the chest, it is evaluated to have a maximum location error of 50%. What is the physical distance of the 50% location error? It is difficult to determine clearly because the reference locations are not marked in Figure 2. Please explain the reference coordinates, the maximum measured distance, and the distance values of the location error.

Author Response

Q1) The authors increased the number of UWB tags from 5 to 9 to evaluate the accuracy of UWB location estimation. When increasing the number of UWB tags, were the tags installed in the order defined by the tag numbers in Figure 2?  

 Yes, clarification has been added in the manuscript. 

Line 176: Anchors were added in increasing order according to Figure 2 

Q2) The locations of Tag 5 and Tag 8 in Figure 2 are unclear. The specific locations of each tag, whether in the living room or the bedroom, need to be indicated.  

Clarification has been added in the Figure label: 

Line 159: Anchors 5 and 8 were located in the living room side of the wall. 

Q3) Figure 3 compares the accuracy according to the location of the UWB anchors. However, the manuscript does not explain how the actual positions were recorded during the experiments and how the exact position information was synchronized with the UWB measurement data. The authors should describe this in detail.  

Clarification has been added in the methodology. Actual locations were measured using a tape measure and marked using masking tape. The coordinates of the bounding box for each location was inputted in a database so that the UWB position could be compared with the real physical location. 

Line 156-159: Furniture-sized locations in the ILS were measured with a tape-measure and each area was marked using masking tape placed on the floor (Figure 2). The respective coordinates of each corner for each location was then used to create virtual bounding boxes for labeling locations that would be used to calculate accuracy of localization. 

Line 181-185: The raw position of the tag was then compared to the preset coordinates of the piece of furniture to calculate accuracy. The data from the ten-second time series was then automatically labeled according to whether or not the coordinate was inside or outside the bounding box. Accuracy was calculated as the proportion of coordinate values in the time series that were within the bounding box as measured by UWB. 

Q4) What are the criteria for the Heading, Pitch, and Roll measurements defined in Table 2?  

Definitions for heading, pitch and roll measurements are have been added. These values were automatically calculated from internal software from Pozyx. Clarification has been added to Figure 1 as well to show X, Y and Z directions on the tag. 

Line 141-143: Euler angles calculated from gyroscope and accelerometry data (Heading as rotation around Z-axis, pitch as rotation around Y-axis and roll as rotation around X-axis in Figure 1) 

Q5) According to Figure 3, when the UWB is located on the chest, it is evaluated to have a maximum location error of 50%. What is the physical distance of the 50% location error? It is difficult to determine clearly because the reference locations are not marked in Figure 2. Please explain the reference coordinates, the maximum measured distance, and the distance values of the location error.  

Thank you for this question. The accuracy was calculated according to if the UWB position was within or outside the bounding box (as in Q3). We did not calculate the physical location errors and felt it would be confusing to the reader if we present location accuracy values in detail. However, from a previous study, we found the mean absolute difference from a point in the ILS and the same position measured from UWB was 27cm. We did not do the same analysis for this study, as we were more interested in classifying if a person was at a piece of furniture or not. We do have the data for the study to do a maximum measured distance analysis, but would require more time for our reviewer response period. 

Round 2

Reviewer 1 Report

Comments and Suggestions for Authors

The authors addressed all the previous comments satisfactorily.